# Vumark's Method of Production Layout Designing

**Juraj Kováč** [1], **Peter Malega** [1], **Vladimír Rudy** [1] and **Jozef Svetlík** [2,*]

1   Department of Industrial and Digital Engineering, Faculty of Mechanical Engineering,
    Technical University of Košice, Letná 9, 040 01 Košice, Slovakia
2   Department of Manufacturing Machinery and Robotics, Faculty of Mechanical Engineering,
    Technical University of Košice, Letná 9, 040 01 Košice, Slovakia
*   Correspondence: jozef.svetlik@tuke.sk; Tel.: +421-55-602-2195

**Abstract:** The paper deals with the issue of the mixed-reality usage in the design of production systems, its changes during expansion, or technological changes in the production, where it is necessary to flexibly and quickly verify the integration of a new machine into the existing layout and eliminate collision situations even before the installation of a physical machine in production. This is realized through Vumark's design methodology, which was verified and applied in the conditions of the production environment of the Innovation and Prototyping Centre in the Faculty of Mechanical Engineering at the Technical University of Kosice. The individual parts of the paper describe the Vumark deployment procedure in the production area and the software and hardware resources that the user can apply. Three production machines and one robotic device selected from the Factory design software database were chosen for the experiment. The chosen method enables us to verify during the experiment several variants of deployment the machines in the layout and thus to find the optimal location in a very short time. The experiment showed that the chosen method is applicable in practice and brings savings of time, costs, and energy especially when changing the layout or replacing the machine in the production hall.

**Keywords:** virtual reality; Vumark; mixed reality; design; marker

## 1. Introduction

Industrial production is a complex process that contains a series of activities, where each activity operates according to certain rules that ensure quality and safe operation. Every quality production has to be supported by the honest work of project teams. The solution and spatial optimization of the layout of the production technology in production is an integral part of designing production systems. In the current design, employees have reliable software and high-quality technical equipment that ensure accurate data and outputs. The paper points out one of the possible solutions for the design of production systems. The modelling of the production system design of machines and equipment is processed in CAD software version 2022.3 with the support of internal libraries, which can be used as a possibility for creating models. Of course, the mentioned method and software are not the only variant of solving the given problem. The result of the experiment is a technology that uses mixed reality in the design of production systems.

At the beginning, it is necessary to understand the overall view of how mixed reality works. A so-called virtual continuum displays a continuous range between complete virtual reality and the real environment. This range includes all possible versions between the proportions of real and virtual elements. The mixed reality is located precisely in the area between the two extremes, i.e., the elements. [1]. Thus, mixed reality actually consists of augmented reality and augmented virtuality according to the parts of virtual elements.

The Industry 4.0, as a challenge of today´s highly competitive environment, supposes that internet technologies will be used in future factories and that they will replace used components by using dynamic and intelligent cyber-physical systems that are the mixture

of the physical objects with their digital representation [2–5]. A research team led by Quint [6] in their paper proposed a system construction for a mixed-reality-based learning environment that gives together the physical objects and visualization of their digital content via augmented reality. It allows making the dynamic interrelations between real and digital factory visible and tangible.

Mixed reality involves the integration of virtual computer graphics into real 3D space, or the real-world elements in a virtual environment [7].

In Coutrix and Nigay's research [8], a new model of interaction for mixed-reality systems was introduced. The main benefits of the mixed interaction model are:

- the unification of several existing approaches to mixed-reality systems such as tangible user interfaces, augmented virtuality and augmented reality, as well as approaches to a more classical graphical user interface, especially the instrumental interaction model
- the study of mixed-reality systems in the intensions of modality and multimodality.

Mixed reality is classically a highly interdisciplinary field that deals with signal processing, computer vision, computer graphics, user interfaces, human factors, wearable computers, mobile computers, information visualization, and display and sensor design [9].

Mixed reality creates a space for the coexistence of physical and virtual elements, which allows the easy interaction between them. This view does not look on the space as two independent areas, where one is prior to another, but it blurs the boundaries between areas and creates bigger space where components from both areas can communicate in real time [10].

The research made by Hughes et al. [11] treats how to transform the core of mixed reality technology and methods into diverse urban terrain applications. This type of research is significant, because these types of applications can be used for military training and situational awareness, as well as for community learning to significantly increase the entertainment, educational, and satisfaction levels of existing experiences in public venues.

A mixed-reality-based user interface for the quality control monitoring of automobile body surfaces is developed in the research made by Munoz et al. [12] in order to increase worker ergonomics and productivity. The results showed that the suggested mixed-reality-based interface improved user ergonomic parameters in terms of comfort, as well as user productivity in terms of time and accuracy when performing quality control tasks.

Authors Lee, Han, and Yang [13] developed a scheme for creating a mixed reality-based digital manufacturing environment in which physical items, such as real images, are grouped with the virtual space of virtual products. Using this technology, they created a mixed-reality-based virtual factory layout-planning system and demonstrated its usefulness in the simulation of process-layout design. Their system is built on the usage of an image-based tracking method that extracts an arbitrary feature from actual pictures, such as a safety sign; such images can be utilized instead of artificial identifiers because they are easily obtainable from the place.

Through a comparison of commercially accessible mixed-reality applications, this study intends to assess the utility of mixed-reality technology in architectural design and construction layout. The study used two HoloLens devices linked to a hard hat to comply with construction safety rules and discovered nine mixed-reality applications designed for architectural and construction objectives. The capabilities of mixed-reality technology in these applications were assessed and compared for architectural design and construction layout under various on-site conditions [14].

The paper wrote by Lee et al. [15] presented a way for implementing mixed reality into the production process, as well as being a method for adapting common security signposts in the plant to replace a black square marker for visual fiducial recognition.

This paper's discussion of the combination of maintenance systems and mixed reality techniques deals with the development of data visualization. The operator will be supported during maintenance activities by the overlap of virtual components in a real scene, allowing the operator to work in a safe and precise manner in the mixed environment of industrial maintenance [16].

Maintenance data and changes to the physical object should be identified digitally in the future and transferred to the data management system in real time. A digital twin should allow for the unambiguous assignment of data to the appropriate elements, resulting in a virtual reflection of the condition of the industrial plant. This paper describes an approach to create a system-based digital twin for an industrial plant's maintenance phase, which will allow data to be transmitted from mixed reality to a Product Data Management System [17].

The goal of the paper made by Kokkas and Vosniakos [18] is to use mixed-reality tools to create machine layouts, particularly for Flexible Manufacturing Systems. The purpose is for the user to assess alternative layouts based on non-measurable characteristics including operator experience, empirical or non-tacit knowledge, and on-site impression. By simulating the whole production process, which includes movement, manipulation, and processing of parts using real and virtual equipment in parallel and serial coexistence, the functional connection between the collaborating real physical and virtual machinery of the layout is enabled. The app is made with the ARKitTM version 5.1.0 API tool and Unity3D 2022.1.0.

The mixed-reality approach to simulation for testing reasons connected to automation software is discussed in this study. The objective is to evaluate real automation SW, such as the MES layer, by mixing real machines or components with simulated ones (in their example, using a 3D graphic simulation tool). Real machining centers, robots, pallets, conveyor belts, PLCs, and CNCs were implemented in a real application scenario. The exchange of products and information between real and virtual components has been realized with the help of a Java-based layer and an appropriate interchange zone [19].

The research paper written by Sautter and Dalling [20] looked at the benefits and drawbacks of mixed-reality-assisted on-the-job learning and described some practical implications for industrial training. The research of two use cases (the first: the training of new employees for the semi-automated assembly lines in the production of pneumatic cylinders revealed that mixed reality, and the second: includes learning modules with a focus on declarative and cognitive knowledge) technologies can assist learning factories overcome their constraints and maximize their potential for successful industrial training in key constructivist learning areas. To get the most out of mixed-reality technologies, they should be used for specific learning objectives and incorporated into a modular didactic framework, taking into account all of the features of successful mixed-reality application.

Using Mixed Reality and Ubiquitous Sensor Networks, this article provides a rapid and effective inspection method to ensure product performance following the design and manufacturing procedure. The addition of 3D information about the product has shown the improvement in workers understanding of the information for proper inspection. Furthermore, the connection of Mixed Reality and Ubiquitous Sensor Networks can assist in the efficient inspection of items because all essential information is available and may be supported anywhere in the system [21].

Lee et al. [22] proposed a method for designing virtual factory layouts based on mixed reality using legacy data that has previously been built in the real world. To do this, this research team created a way of acquiring simulation data from legacy data and processing it for mixed-reality visualization. They also built a mixed-reality display system that can recreate a virtual manufacturing layout using processed data. By validating the position and application of equipment in advance before arranging real ones on the project site, the developed method can eliminate industrial layout errors.

Production plants have faced numerous obstacles in recent years, notably those related to variable demand and changing customer and supplier requirements, forcing new technical roadmaps and interventions in production processes. Innovative technologies support the development of information processes aimed at employees in this proactive context. With this fact, augmented and virtual reality can be used for workforce training; they should be able to effectively engage with a human workforce [23].

The proposed mixed reality structure consists of five layers: the first layer considers system parts; the second and third layers concentrate on architectural issues for component integration; the fourth layer is the application layer that executes the architecture; and the fifth layer is the user interface layer that allows user interaction [24].

During the COVID-19 pandemic, the suggested model with the usage of mixed reality provides a comprehensive solution for huge building complexes and industrial parks, guaranteeing public safety and also the health and well-being of the facilities management team. The mixed reality technique allows for the remote processing of path layouts, avoiding human interaction and ensuring that there is no chance of virus transmission [25].

In mechanical engineering, there are possibilities to use not only CL data for the machining process, but it also shows the possibilities of using simulation data from the pre-production phase for augmented reality. This research links the techniques used in the educational process with the necessities of real practice and a direct visualization solution for the operator in the machining process is also proposed [26].

## 2. Materials and Methods

When we want to design the layout of production and distribution of markers, it is necessary to observe certain minimum distances between individual production equipment and fixed elements of building structures (e.g., pillars, walls, etc.) [27].

1.  A raster network model is assigned to the reference space for the location of the production system, which is dimensioned via the capacity calculation of the required areas, volumes, and production means.
2.  According to the production (functional) activity, or production optimization, that has to take place in the reference space, work zones, resp.; sub-zones of the production process are placed in the raster, and the location of the Vumark is selected to display the virtual model in a real environment using a helmet HoloLens 2 for mixed reality. In the case of absence of a helmet for mixed reality, it can be used a smartphone or tablet that support virtual reality.
3.  Models of topological relations of construction elements of the production system structure (already existing production machines, virtual production machines, handling and auxiliary equipment, etc.) are solved in individual zones which can also be included in Vumark and displayed then in the mixed reality.
4.  Computational-interactive way of solving topological relations is realized on the basis of using simplified 2D models, accurate 3D models of construction elements of the production system, resp. These 3D models of production machines can be obtained from databases of software products or can be modeled using CAD systems at a 1:1 scale to match the size of a real, physical production machine when these production machines will be displayed in a real environment.
5.  From the graphic database of CAD system are selected specific 2D, 3D models of the relevant building elements to the calculated reference points of their location, resp. In the layout of the production system, the raster model will be displayed on the floor in the real environment using a mini projector placed on the structure at the appropriate height, and the exact location of the Vumark will be defined.
6.  Based on interactive decision-making procedures and solution visualization, the optimal variant of the production system structure is selected. In the case of a collision situation between the machines, the placement of the Vumark will be optimized.

The basis of the solution is a model of topological relationships of a pair of construction elements of the production system structure (Figure 1). Based on the usage of the combinatorial principle, it is possible to create various variants of more complex structures of production systems from the basic model (Figures 2–4).

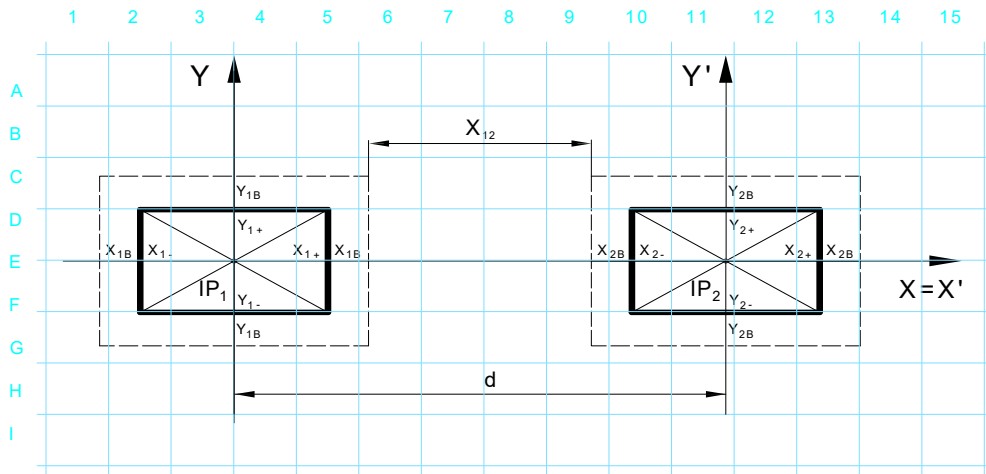

**Figure 1.** Model of building elements relationships of production systems 1 - $0_x$ [28].

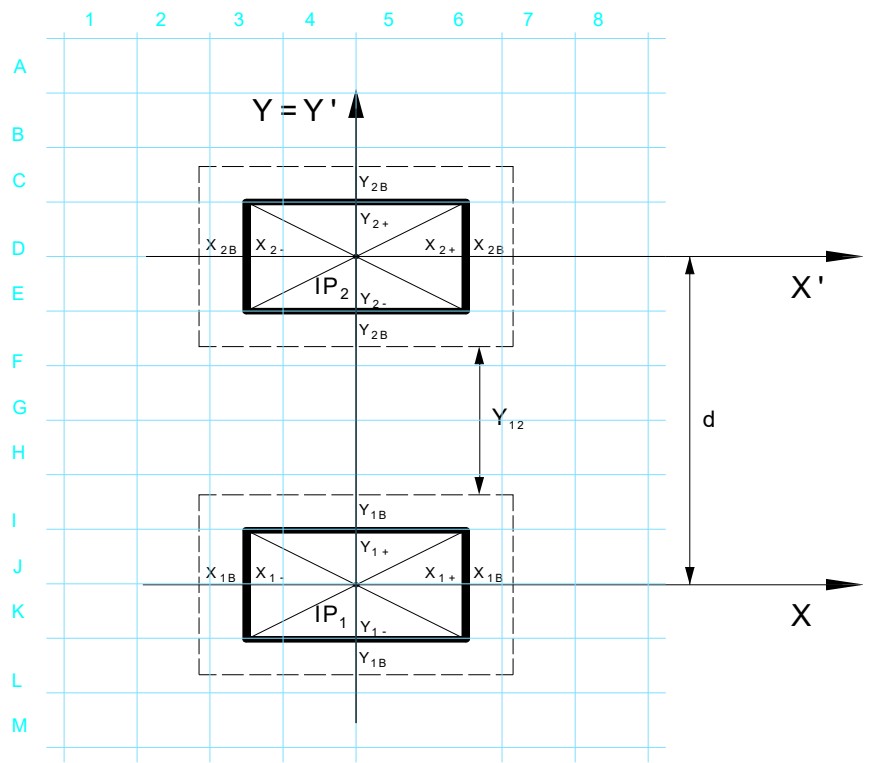

**Figure 2.** Model of building elements relationships of production systems 1 - $0_y$ [28].

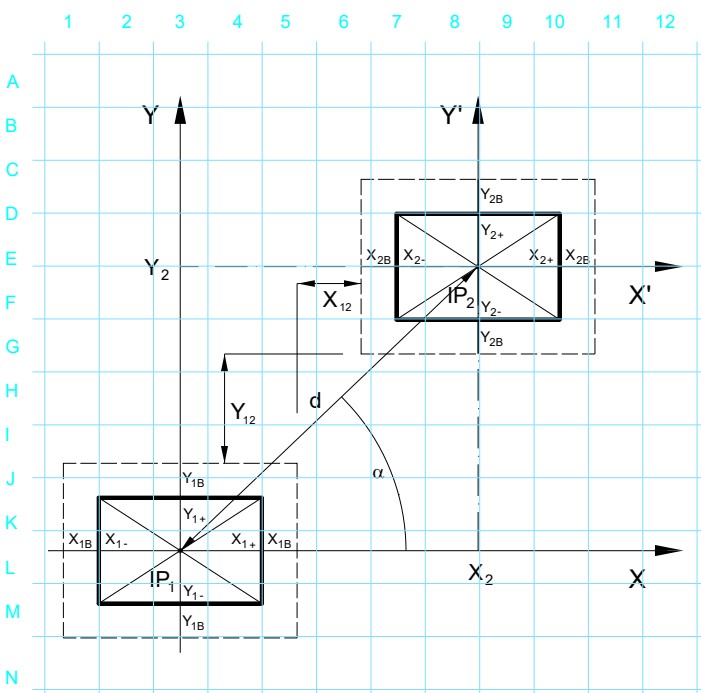

**Figure 3.** Model of building elements relationships of production systems 1 - $\alpha$ [28].

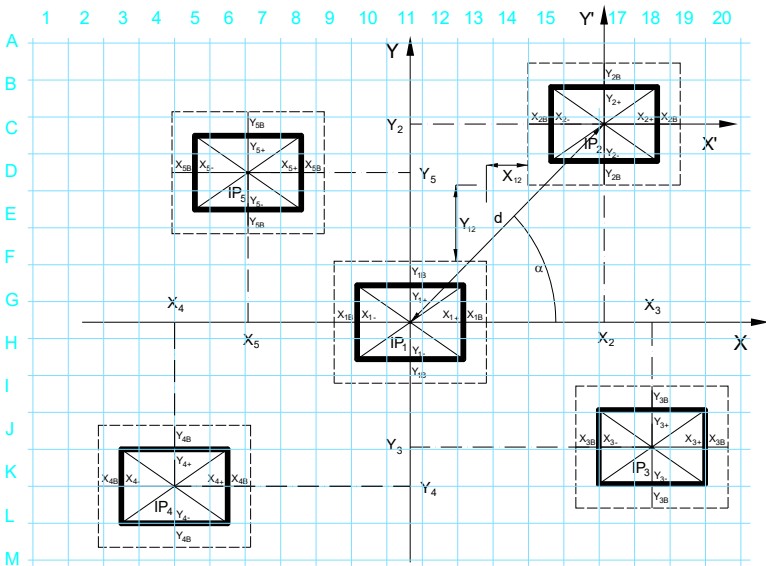

**Figure 4.** Relationship model for pilot structures of production systems [28].

Syntax for the designation of parameters of the topological model of located construction elements of the production system are as follows [28]:

$x_{i+}$ dimension in the positive direction of the x-axis, $x_{i-}$ dimension in the negative direction of the x-axis,

$y_{i+}$ dimension in the positive direction of the y-axis, $y_{i-}$ dimension in the negative direction of the y-axis,

$x_{iB}$, $y_{iB}$ parameters of the safety zone, $x_i$, $y_i$ coordinates of the insertion point,

$x_{ij}$ distance between the *i*-th and *j*-th object in the direction of the x-axis,

$y_{ij}$ distance between the *i*-th and *j*-th object in the direction of the y-axis,

*i, j* serial number,

d distance of insertion points,

$\alpha$ angle of rotation.

The insertion point of the construction element of the production system is located on the $0_x$ axis.

$$IP_1 = [x_1, y_1, 0] = [0, 0, 0], IP_2 = [x_2, y_2, 0], \alpha = 0 \tag{1}$$

$$x_2 = x_{1+} + x_{1B} + x_{2B} + x_{2-} + x_{12}, \; y_2 = y_1 = 0 \tag{2}$$

$$d = x_{1+} + x_{1B} + x_{2B} + x_{2-} + x_{12} \tag{3}$$

$$x_{12} = d - (x_{1+} + x_{1B} + x_{2B} + x_{2-}) \tag{4}$$

$x12 > 0$ the condition must be met in order to avoid collisions of the distributed elements.

The insertion point of the construction element of the production system is placed on the axis oy:

$$IP_1 = [x_1, y_1, 0] = [0, 0, 0], \; IP_2 = [x_2, y_2, 0], \alpha = \frac{\pi}{2} \quad x_2 = x_1 = 0 \tag{5}$$

$$y_2 = y_{1+} + y_{1B} + y_{2B} + y_{2-} + y_{12}, y_{12} = d - (y_{1+} + y_{1B} + y_{2B} + y_{2-}), \\ Y_{12>0} \tag{6}$$

The condition must be met in order to avoid collisions of the distributed elements.

The insertion point of the construction element of the production system is rotated by an angle $\alpha$, of the flow line of the insertion points and the axis $0_x$.

$$IP_1 = [x_1, y_1, 0] = [0, 0, 0], \; IP_2 = [x_2, y_2, 0], \; \alpha \in \left(0, \frac{\pi}{2}\right), \; \alpha = arctg \frac{y_2}{x_2} \tag{7}$$

$$x_2 = x_{1+} + x_{1B} + x_{2B} + x_{2-} + x_{12}, y_2 = y_{1+} + y_{1B} + y_{2B} + y_{2-} + y_{12} \tag{8}$$

$$d = \sqrt{x_2^2 + y_2^2} = \sqrt{(x_{1+} + x_{1B} + x_{12} + x_{2B} + x_{2-})^2 + (y_{1+} + y_{1B} + y_{12} + y_{2B} + y_{2-})^2} \tag{9}$$

$$x_{12} = x_2 - (x_{1+} + x_{1B} + x_{2B} + x_{2-}) \quad y_{12} = y_2 - (y_{1+} + y_{1B} + y_{2B} + y_{2-}) \tag{10}$$

$$x_{12} \geq 0 \vee y_{12} \geq 0 \quad (x_{12} \neq 0 \wedge y_{12} \neq 0) \tag{11}$$

The conditions must be met in order to avoid collisions between the distributed construction elements of the system (the geometric characteristics of individual production systems are also important for modelling the spatial relationships in the production systems. It is necessary to take into account spatial relations and basic geometric characteristics: outer shape and dimensions, maximal dimensions of the working space, orientation and position of the coordinates in the coordinate system of the production element with respect to the reference-coordinate system and geometric characteristics of the working space).

If the first insertion point is placed at the beginning of the coordinate system, then there are four possibilities when inserting the second point due to the division of the plane into quadrants. The insertion point can be located in:

- Quadrant I.

$$\alpha \in \left(0, \frac{\pi}{2}\right) \tag{12}$$

- Quadrant II.

$$\alpha \in \left(\frac{\pi}{2}, \pi\right) \tag{13}$$

- Quadrant III.

$$\alpha \in \left(\pi, \frac{3\pi}{2}\right) \tag{14}$$

- Quadrant IV.

$$\alpha \in \left(\frac{3\pi}{2}, 2\pi\right) \tag{15}$$

For Quadrant I (if the inserted point is at the top right of the base element):

$$IP = [x, y, 0]$$

$$x_2 = x_{1+} + x_{1B} + x_{2B} + x_{2-} + x_{12} \tag{16}$$

$$y_2 = y_{1+} + y_{1B} + y_{2B} + y_{2-} + y_{12} \tag{17}$$

For Quadrant II. (if the inserted point is at the top left of the base element):

$$IP = [x, y, 0]$$

$$x_5 = -(x_{1-} + x_{1B} + x_{5B} + x_{5+} + x_{15}) \tag{18}$$

$$y_5 = y_{1+} + y_{1B} + y_{5B} + y_{5-} + y_{15} \tag{19}$$

For Quadrant III. (if the inserted point is at the bottom left of the base element):

$$IP = [x, y, 0]$$

$$x_4 = -(x_{1-} + x_{1B} + x_{4B} + x_{4+} + x_{14}) \tag{20}$$

$$y_4 = -(y_{1-} + y_{1B} + y_{4B} + y_{4+} + y_{14}) \tag{21}$$

For Quadrant IV. (if the inserted point is at the bottom right of the base element):

$$IP = [x, y, 0]$$

$$x_3 = x_{1+} + x_{1B} + x_{3B} + x_{3-} + x_{13} \tag{22}$$

$$y_3 = -(y_{1-} + y_{1B} + y_{3B} + y_{3+} + y_{13}) \tag{23}$$

The following applies to the size of the angle $\alpha$ (angle of rotation IP2):

$$tg\alpha = \frac{y_2}{x_2} \rightarrow \alpha = arctg\frac{y_2}{x_2} \tag{24}$$

When we insert elements into individual quadrants, inequalities for x- or y-coordinates must be met in order to maintain safety distances.

$$x_2 - x_1 \geq x_{1+} + x_{B1} + x_{B2} + x_{2-} \quad x_3 - x_1 \geq x_{1+} + x_{B1} + x_{B3} + x_{3-} \tag{25}$$

$$|x_4 - x_1| \geq x_{1-} + x_{B1} + x_{B4} + x_{4+} \quad |x_5 - x_1| \geq x_{1-} + x_{B1} + x_{B5} + x_{5+} \tag{26}$$

$$y_2 - y_1 \geq y_{1+} + y_{B1} + y_{B2} + y_{2-} \quad |y_3 - y_1| \geq y_{1+} + y_{B1} + y_{B3} + y_{3+} \tag{27}$$

$$|y_4 - y_1| \geq y_{1-} + y_{B1} + y_{B4} + y_{4+} \quad y_5 - y_1 \geq y_{1+} + y_{B1} + y_{B5} + y_{5-} \tag{28}$$

Models of typological relationships in production systems are characterized by considerable universality in terms of application conditions, as they are suitable for optimization of the production system design, but also for the traditional design of production systems.

The typological relationships presented in the paper are based on generally valid spatial principles and methodologies that are obligatory when the layout of production equipment on the production area is solved. These are applied in decision-making software algorithms. Their output is the optimal solution that eliminates the conflicting states of the location of the new means of production related to the immediate surroundings of the new layout, which is located through the Wumark brand and its optimal location is transformed into the mixed reality.

When choosing the location of the Vumark, we used a projector to project the raster on the surface of the production hall, with which we determined the exact location of the production equipment, and we eliminated collision situations when inserting the production equipment into the layout of the production system.

On the basis of the mentioned procedure, it is possible to place Vumark in the layout solution within the optimization of production system and classic mechanical engineering productions for piece and small-batch production.

The mentioned mathematical apparatus can be used in the placement of production equipment in shapes that are often used in such types of production (e.g., type "U", type "L", type "I", type "diagonal" and type "star").

## 3. Results and Discussion

### 3.1. Mixed Reality in the Design of Production Systems

Augmented reality tools can be used effectively in the design of production and logistics processes. Today, computer technology and its other options are an integral part of the design of production systems. The fact that the digital model of the production system can be easily adopted into a real environment is a big advantage of augmented reality for designers [28].

The principle lies in the placement of markers in the production space and the assignment of virtual objects to individual signs. Using head-mounted display or stereoscopic 3D glasses, the designer can view the proposed layout. Objects can be created in various software applications, such as AutoCAD, CATIA, Solidworks, etc. and are displayed in scale 1:1, which allows a realistic view of the future production layout. The main benefits are the reduction of risk while introducing new production, improvement of the location of production machinery and equipment, reduction of the required area, detection of potential collisions, and better understanding about the proposed solution. Visualization of a 3D-production system model in a real environment is in Figure 5.

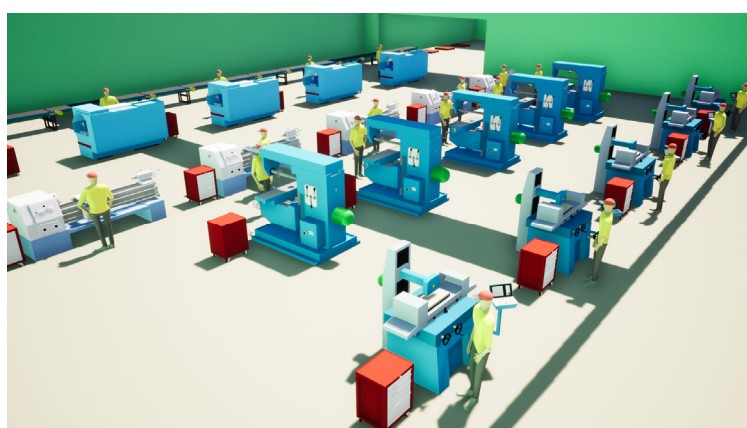

**Figure 5.** Visualization of a 3D production system model in a real environment.

From the several options for displaying models using AR, this method is first-class, but more time-consuming and software-intensive. Another possibility is to use an interactive planning table, in which we will implement the environment in which we want to display the models [1]. Augmented reality in this case complements the 2D drawing of the layout displayed on the touch screen area of the planning table with a 3D model of the production system. It is displayed at the same scale as the 2D production layout (Figure 6). The design-planning table is equipped with software for the design and optimization of the layout solution (e.g., Factory Design). It allows for the making of effective changes in the spatial arrangement of the system, monitoring the impact of changes on the output parameters of the production layout (e.g., length of material flows, total transport performance and transport costs, usage of surfaces, etc.) and at the same time visualizing all changes in the 3D environment using the means of augmented reality [9].

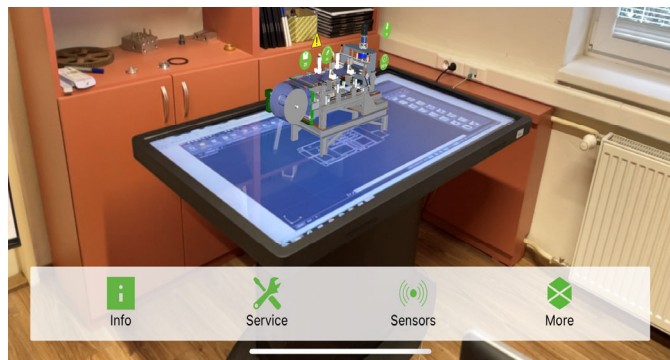

**Figure 6.** Visualization of a 3D model using drawing documentation.

Production assets can be viewed using:

- 2D-drawing documentation, compatible with software (PC, laptop, mobile device, tablet),
- 3D glasses,
- Directly using a mobile device or tablet.

### 3.2. Software Equipment for Mixed Reality

Software from Autodesk was used to create models of production means within the experiment. Software libraries contain production machines and devices that are divided according to usage and have their own specific names, so it is easy to find them in the system. These libraries offer customized devices that the user can easily find and download directly from the library to the software.

Autodesk Inventor Professional software version 2022.3 offers the support of the Asset Browser library. It is necessary to install the product Autodesk Factory Design Utilities, which provides a library with a wide range of production machines and equipment (Figure 7).

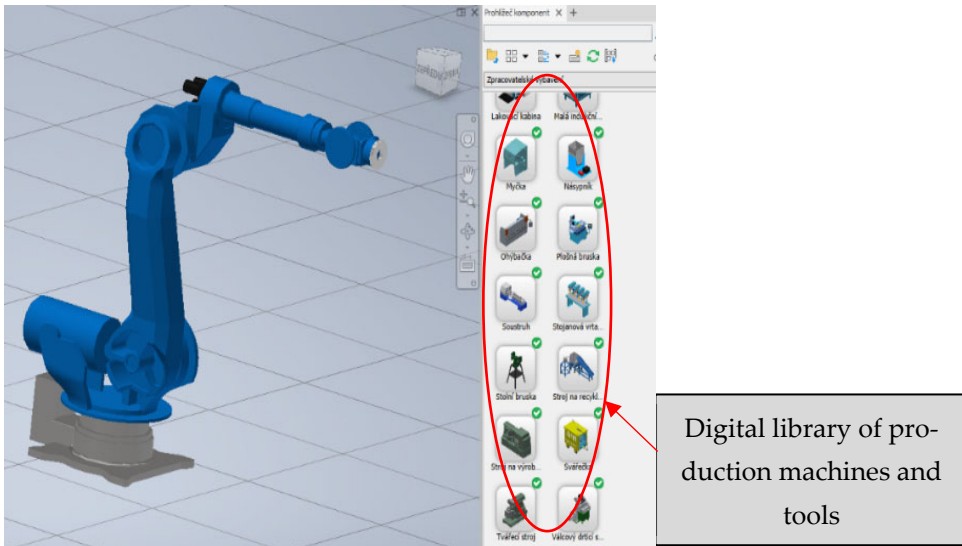

Digital library of production machines and tools

**Figure 7.** Visualization of a digital library of production equipment.

To verify the mixed reality in the software, it is necessary to use the mobile application Vuforia View, which, however, is not available for all devices. When a potential user buys a mobile device, it is necessary to find out if the smartphone or tablet has mixed-reality support.

The procedure for working in Vuforia Studio is as follows:

- Export of the model in Inventor in the correct CAD format.
- Creation of a project in Vuforia and set of the project properties.
- Import of the CAD model into Vuforia Studio.
- Work with the model, calibration of the model, and creation of Markers.
- Saving, publishing, and printing of Markers.

If the model is modelled as the configuration in the Vuforia software version 9.8.0, it is possible to mark the components from which the model is created and with each component, it is possible to work separately. If the model is created as one part, then the software cannot recognize parts of the model. View of the model in Vuforia Studio is in Figure 8.

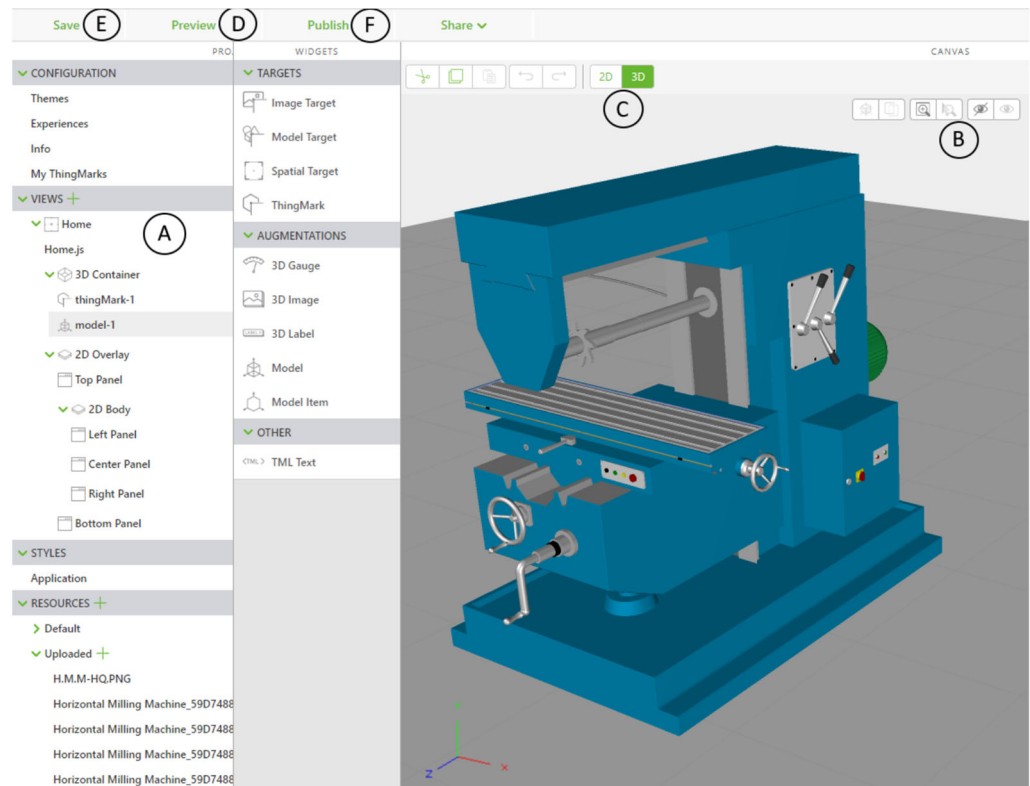

**Figure 8.** View of the model in Vuforia Studio [29].

### 3.3. VuMark Printing and the Vuforia View Mobile Application

ThinkMark is a type of barcode, and it is one of the leading trends in the design while coding data for AR. ThinkMark is also called a marker. The marker has unique design technology and therefore each marker is unique for the object that can be loaded with it. The advantage of the marker is that it can present an unlimited number of models. It is compatible with various types of formats. The mark is loaded according to its position on the surface. VuMark parts are in Figure 9.

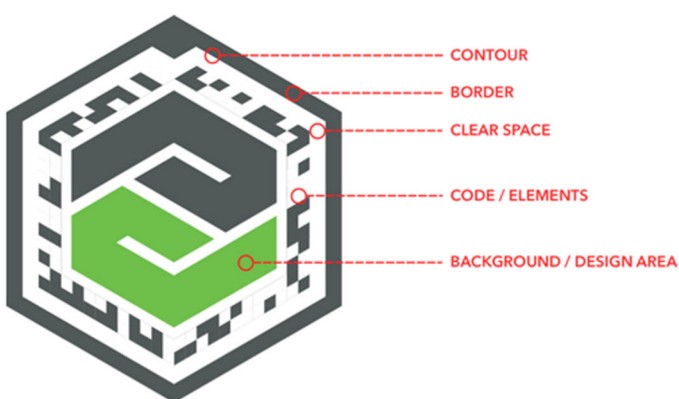

**Figure 9.** Design division of the Marker [30].

*3.4. Own Experiment of Production Layout Design using Mixed Reality Benefits for the Practice*

The "Vumark design" methodology has been verified and until now is still applied in the conditions of the production environment of the "Prototyping and Innovation Centre" operated under the patronage of Faculty of Mechanical Engineering, Technical University of Kosice. The Centre, through its activities, ensures the cooperation with the production sector and participates in the cooperation that lead to product and technological innovation. In its technological configuration are top CNC production equipment, 3-5-8 axis machining centers and laser technologies for precise production of components. The majority of the production technique consists of the equipment from the company DMG Mori Brno. From the system of production to the order of operating, the Centre requires regular expansion, or changes in its technological capabilities. This fact places increased demands on the reconfiguration and flexibility of the production environment within not only the operating hall, but also calls for reconfiguration changes between the operated objects of the Centre (hall PK12A, hall PK14). The necessity for a spatial reorganization of the production structure of the Centre is caused by a new customer request. Its implementation on the existing configuration of the production space depends primarily on the volume of production, the dimensional size of the production objects, or the necessity to incorporate new technology into their production. The preparation and physical implementation of such operational reconfiguration projects would be time consuming in the classical approach. In addition, such spatial reorganization of production technology does not always have to be the optimal one. In addition, its timely implementation may threaten the fulfillment of contractual conditions, or in terms of time, it is partially realized at the expense of the production times of physical production. The solution of such a quick reconfiguration task is based on the principles of "Vumark's design methodology" in the application of virtual-reality elements. The production hall of the PK12 prototype innovation center is shown in Figure 10.

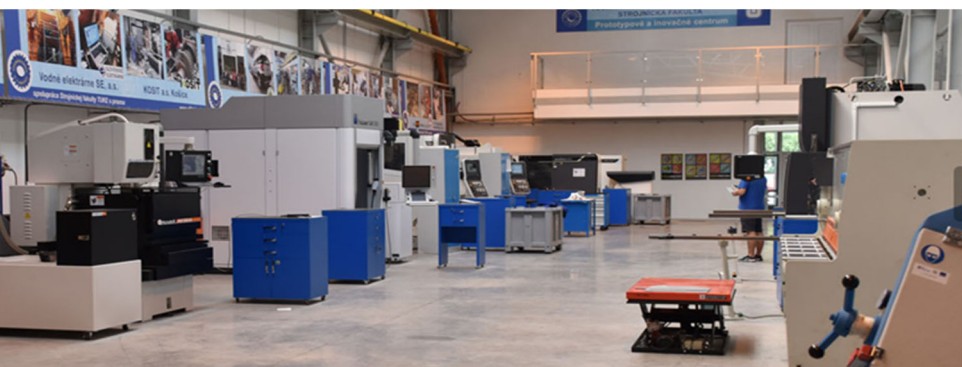

**Figure 10.** A view of the technological part of the production environment of the innovation center.

In the production systems designing using mixed reality as part of the experiment, the preferable solution is a hall with a flat surface, a high ceiling, and sufficient lighting. In the first step, it is important to place the printed Markers in the area that is intended for the machine position. Placement of markers in space is in Figure 11. In our case, we used the Prototype and Innovation Center at the Technical University in Košice for the experiment, where we placed Machine Markers in the existing layout and through the used Hololenz helmet for mixed reality; we visualized the production machines in a real environment. The production machine can be displayed from all angles and perspectives, and it is possible to eliminate any collision situations that can occur in the real production process. In addition to the helmet for mixed reality, it is possible to use a tablet or smartphone to display virtual models and through cameras, it is possible to record the placed Marker on the floor. Through Markers, it is possible to design and visualize the complete layout of the production system in the real environment. Production machines or robotic systems can also be moved in the mixed reality and verify the path of movement (e.g., robotic arm) and verify the range of motion, or simulate work on the production machine.

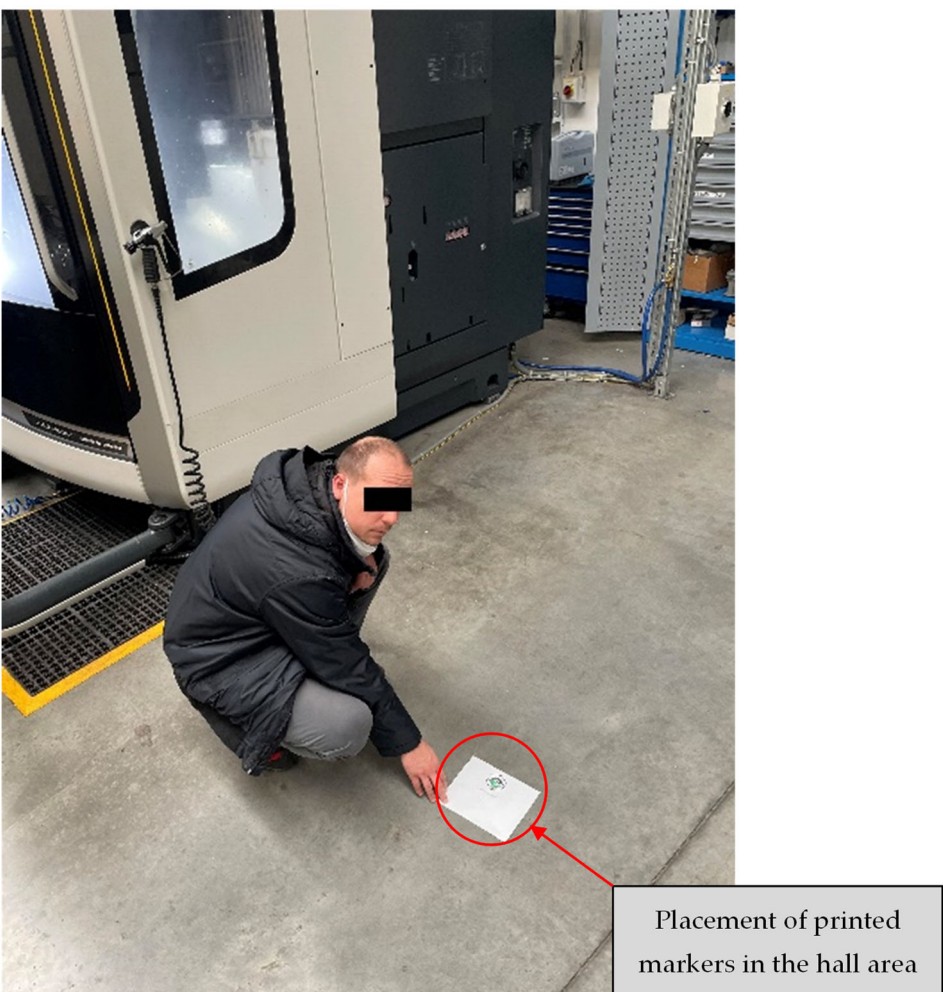

Placement of printed markers in the hall area

**Figure 11.** Placement of markers in space.

Subsequently, it is necessary to scan the markers using a tablet, smartphone, or glasses for mixed reality with the Vuforia View application. The models of production machines shown in this way in mixed reality will show the real area that the machine needs in a reality, which can help in the design of the production system and avoid, for e.g., the collision of machines, or its incorrect placement in the layout of the workplace. The example of the production-machine visualization in the mixed reality is in Figure 12.

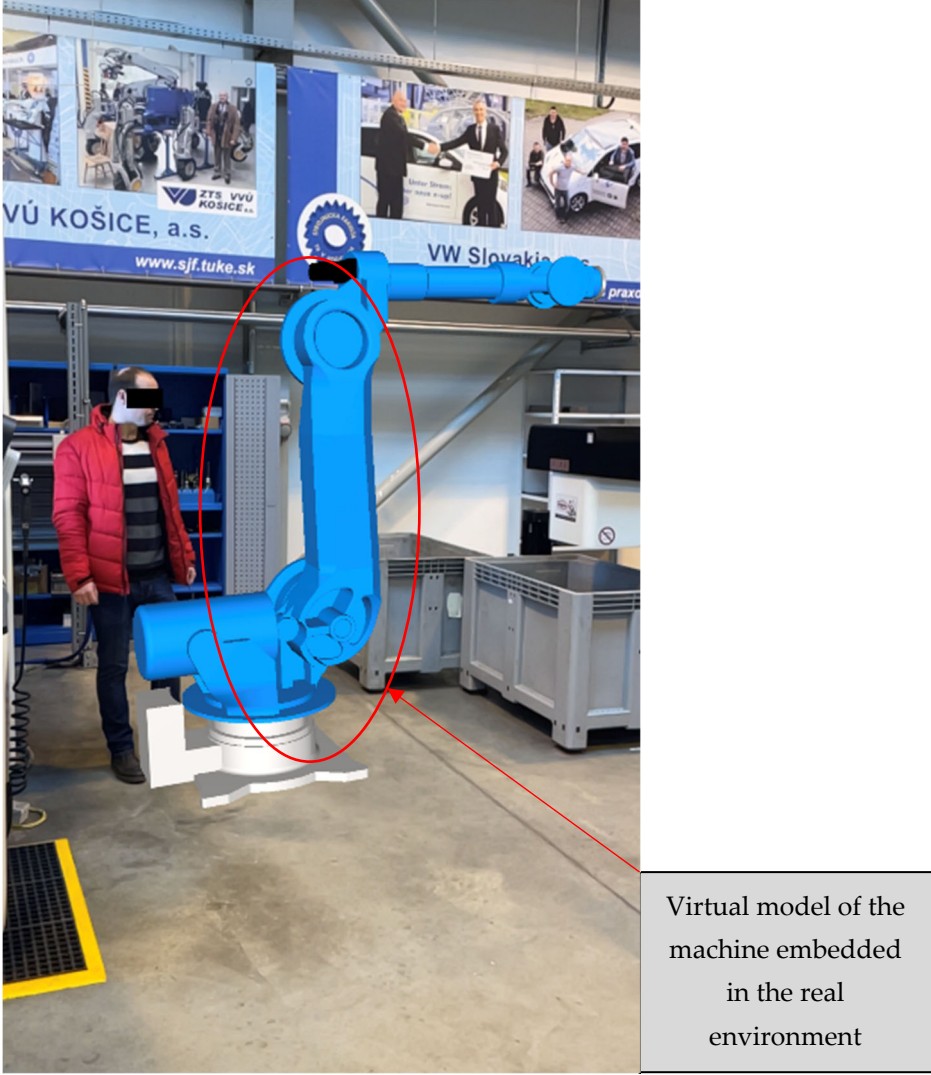

**Figure 12.** Display of the digital machine model in the real environment.

If the digital model of the production machine is modelled as the configuration, it is possible to simulate in a mixed reality the work process of the machine control within the training. The operator may not have a real physical machine but can verify the work progress on the digital model in mixed reality (Figures 13–15).

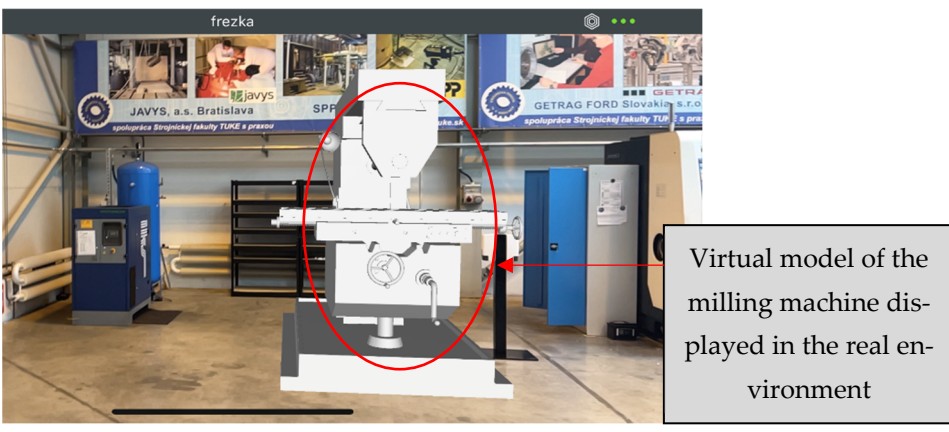

**Figure 13.** Display of the digital model of the milling machine in the real environment.

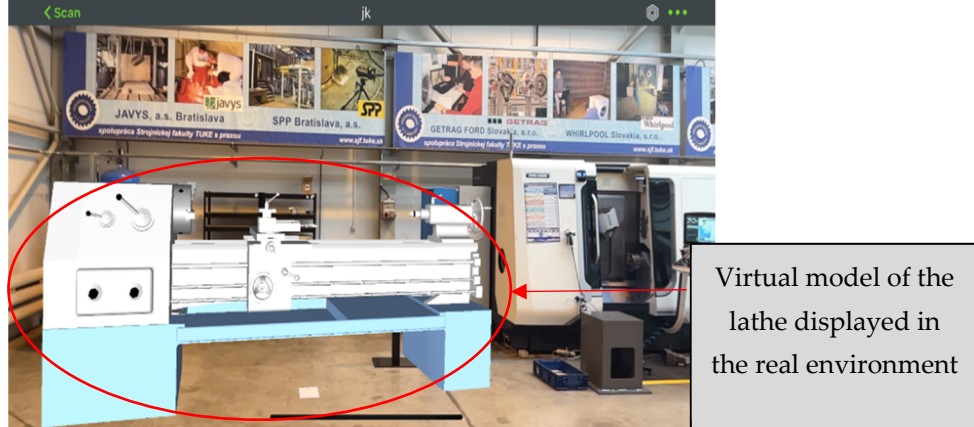

**Figure 14.** Display of the digital model of the lathe in the real environment.

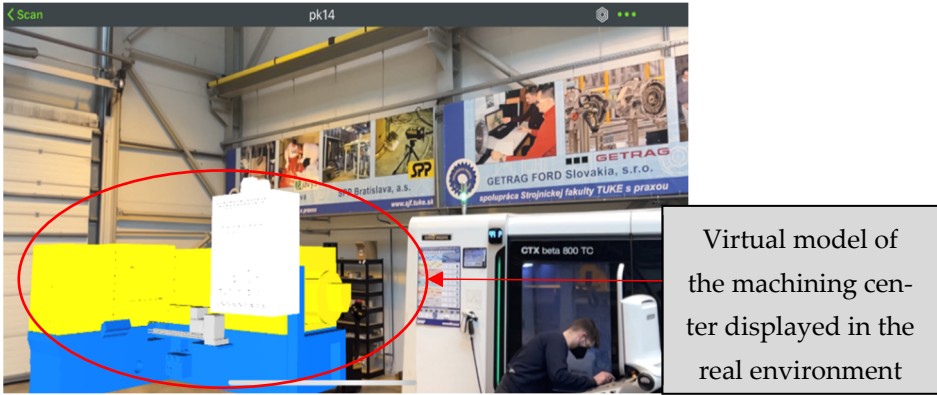

**Figure 15.** Display of the digital model of the machining center in the real environment.

Vumark's design method enables the making of quick changes of various variants when placing machines in the already existing layout of the production system. Using this method, it is possible to immediately detect collision situations between a physical (real) machine and a virtual machine in the production system and thus avoid high costs and time losses during its physical installation. When verifying the placement of the virtual machine in the layout, it is not necessary to shut down the production process as a whole, which also eliminates the economic losses of the production plant. Virtual machine models can be obtained in software such as Factory Design, vis TABLE, and Tarakos, or they can be modelled in a CAD system and implemented in mixed reality very efficiently and in a short time.

The scientific contribution of the documented project methodology is its uniqueness in the given segment of the scientific discipline. From the available literature analyses and surveys it is not possible to find a more sophisticated projection methodology. The considerable advantage of this projection method based on the principles of mixed reality is confirmed by the verification of results in the real conditions of production practice. The time characteristics of its application, when solving the deployment tasks of production techniques of new or of configured productions, fully reflects the preservation of the competitiveness (its increase) of the given production grouping in the dimensions of quick time applicability and the correction of risk factors of the application of the solution verified using simulation elements. The flowchart of the virtual model application to the real environment is shown in Figure 16.

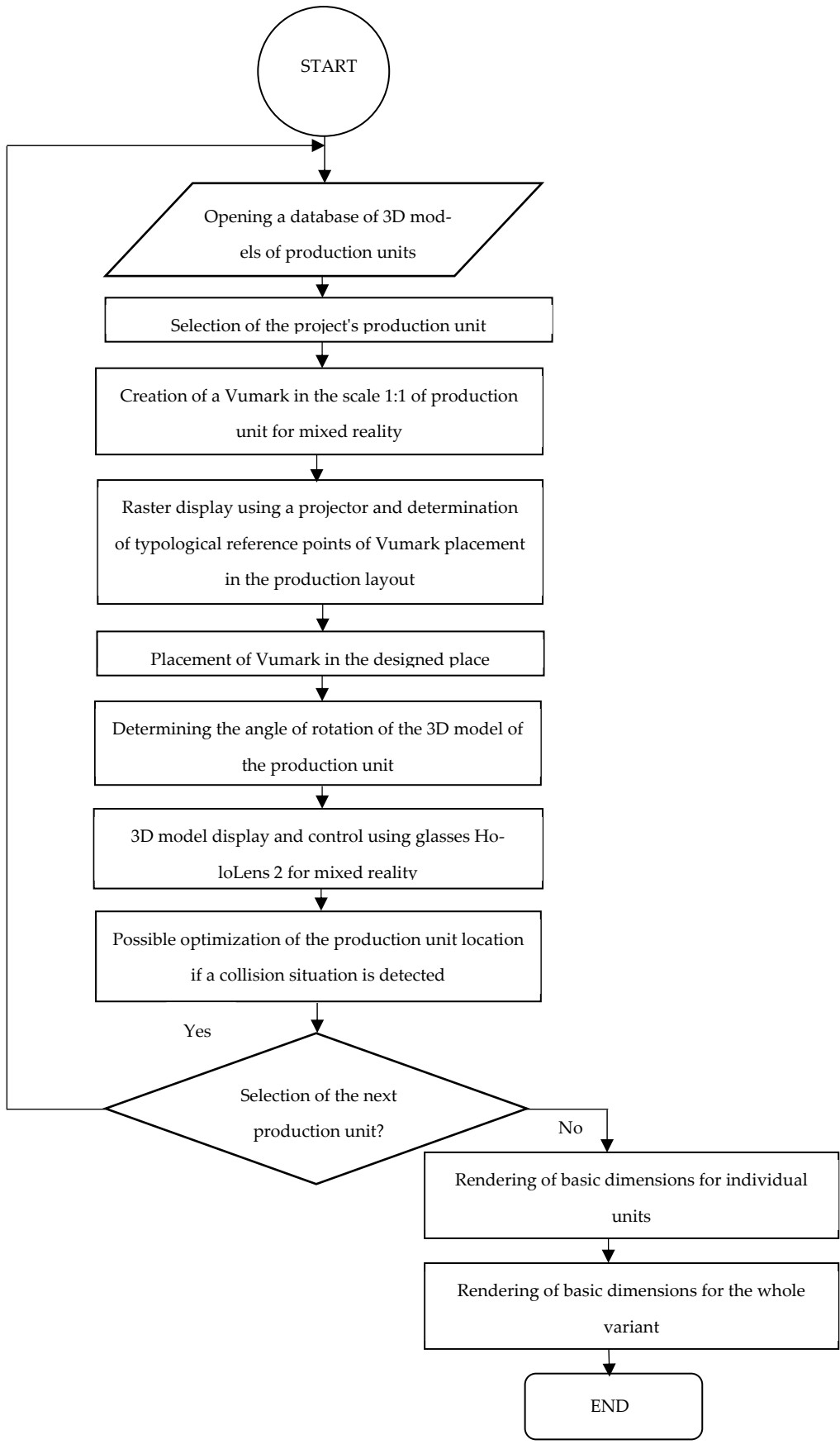

**Figure 16.** Flowchart of application the virtual model to the real environment.

## 4. Conclusions

Currently, the requirements for project activity and the understanding of the new methods development of production systems designing are changing significantly. The system development of design methods, procedures, and techniques is determined by the knowledge of other scientific disciplines, professional creativity, collective decision-making and other factors resulting from market conditions. In designing, the condition of comprehensive knowledge of the problems and their detailed analysis is essential, because it allows for the elimination of the generation of outdated and non-complex approaches of production systems. It leads to the application of new innovative solutions that guarantee the implementation of advanced technologies of the future. Only this approach is the guarantee of identification, optimization, or elimination of project deficiencies even before their future physical implementation [31].

Virtual technologies enter the new areas of application that will significantly change the character of peoples' lives. The improving digitization and virtualization of production and design leads to significant changes in the existing methods of designing production layouts. In the industry sector and especially in the production processes, the real and virtual worlds are increasingly integrated. The share of advanced technologies in all spheres of human activity is increasing. The system of knowledge and methodological procedures obtained from experiments with the system of integration of the virtual and real world shows that the usage of advanced technologies has its importance from a professional and practical point of view. In projects where mixed reality is used to make a visualization and simulation of the dynamic behavior of technical objects or project processes and systems in a real environment, virtual technologies are a significant contribution. This direction of research implemented in the design of production systems is suitable for small and medium enterprises (SME). For the solution of such project activities, from a technical point of view, it is possible to use various types of devices, such as helmet for mixed-reality Hololens 2, etc.

This paper describes an experiment using the Vuforia Studio software. Using Vumark´s digital machine models, which are modeled using a CAD system on a 1:1 scale, these digital models were integrated into the real environment as part of the layout design. The accuracy of the displayed model and technical and display units were tested in the conditions of the production environment of the Innovation and Prototyping Centre operated under the auspices of the Faculty of Mechanical Engineering, Technical University of Kosice.

The main contribution of the paper is the proposal of the methodical procedure for the design of production structures using mixed reality based on the general principles of innovative design. Synthesis conditions for the implementation phase of designing production facilities in mixed reality and experimental use were defined. The method of typological relations was chosen for the implementation of proposals of the layout of production machines and robots and the selection of the optimal variant. The result of this method of project activity in spatial solutions of production systems is, for the necessities of practice, more accurate descriptions of the layout of machines in space using mathematical and other principles. Industrial grids enable the creation of immediate, direct dimensional relationships between the exact surface and the location of the production system that realizes the production process, and the implementation of mixed reality allows us to display a 3D model of the production equipment directly on the selected surface and immediately detect collision situations directly on the place and propose the optimization of the location.

Production systems designing is an area in which it is necessary to follow trends and current opportunities that provide quality solutions and innovative technologies. Mixed reality is one of the technologies, where the potential of usage also has its place in project activities. The results of the experiment show that for a wider application of this projected methodology in practice, it will still be necessary to resolve any problems in detail, such

as the accuracy of displaying objects, their orientation, spatial situation, and the optimal observer distance from the placed object.

**Author Contributions:** J.K. wrote article and created a model within the experiment. V.R. draft theoretical problem. P.M. lead discussion. J.S. made theoretical formulas. All authors have read and agreed to the published version of the manuscript.

**Funding:** This work was supported by KEGA 019TUKE-4/2022 and KEGA 002TUKE-4/2020. This work was also supported by the Slovak Research and Development Agency under the Contract no. APVV-19-0418, APVV-17-0258 and APVV-18-0413.

**Institutional Review Board Statement:** Not applicable.

**Informed Consent Statement:** Not applicable.

**Conflicts of Interest:** The authors declare no conflict of interest. The founding sponsors had no role in the design of the study; in the collection, analyses, or interpretation of data; in the writing of the manuscript, and in the decision to publish the results.

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
