# Peer review of "Vumark’s Method of Production Layout Designing"

_applsci, doi:10.3390/app13031496_

Round 1

Reviewer 1 Report

The article deals with the use of augmented reality in the field of production layout designing. The article is presented in a clear, understandable and transparent way. I consider the main shortcoming of the article to be the lack of description of the actual scientific contribution to the above mentioned field.

Main comments on the article:

1. In the Materials and Methods chapter, the mathematical apparatus for describing the topological relationships between a pair of elements of a manufacturing system is described. The article does not explain how this mathematical apparatus was applied to the designing of the manufacturing layout. How has the mathematical apparatus been implemented in a manufacturing layout design methodology using augmented reality? Can the mathematical apparatus be applied to any of the methods for optimizing the spatial layout of a manufacturing system?

2. The methodology for applying augmented reality for manufacturing layout design is described in a rather general way. Please describe in detail the methodology proposed by the authors in Chapter 3 and in Chapter 5 (Discussion and Conclusions) elaborate on its benefits and possible shortcomings compared to existing approaches.

3. Remove formal errors in the paper:

- wrong numbering of chapters (chapter 4 is missing)

- poor alignment of numbering of mathematical formulas

Author Response

We corrected our paper according to your suggestions. All new aor corrected information are marked in yellow.

  1. In the “Materials and methods” chapter we have added a description of the mathematical apparatus application in the design of the production layout.

      Models of typological relationships in production systems are characterized by considerable   

      universality in terms of application conditions, as they are suitable for optimization of the  

      production system design, but also for the traditional design of production systems.

      The typological relationships presented in the paper are based on generally valid spatial principles 

      and methodologies that are obligatory when it is solved the layout of production equipment on

      the production area. These are applied in decision-making software algorithms. Their output is

      the optimal solution that eliminates the conflict-ing states of the location of the new means of

      production related to the immediate surroundings of the new layout, which is located through

      the Wumark brand and its optimal location is transformed into the mixed reality.

      When choosing the location of the Vumark´s, we used a projector to project the raster on the

      surface of the production hall, with which we determined the exact loca-tion of the production

      equipment and we eliminated collision situations when insert-ing the production equipment into

      the layout of the production system.

      On the basis of the mentioned procedure, it is possible to place Vumark´s in the layout solution

      within the optimization of production system and classic mechanical engineering productions for

      piece and small batch production.

     The mentioned mathematical apparatus can be used in the placement of produc-tion equipment

      in shapes that are often used in such types of production (e.g. type “U”, type “L”, type “I”, type

     “diagonal” and type “star”).

  1. In the “Result and Discussion” chapter, the method of mixed reality application in designing the layout of the production system was described in more detail, and in the “Conclusion” chapter, the advantages of the described method application regarding the design of the production layout and its advantages were added.

      Vumark's design method enables to make quick changes of various variants when placing   

      machines in the already existing layout of the production system. Using this method, it is possible

      to immediately detect collision situations between a physical (re-al) machine and a virtual

      machine in production system and thus can avoid high costs and time losses during its physical

     installation. When verifying the placement of the virtual machine in the layout, it is not necessary

     to shut down the production process as a whole, which also eliminates the economic losses of the

     production plant. Virtual machine models can be obtained in software such as Factory Design, vis

     TABLE and Tarakos, or they can be modelled in a CAD system and implemented in mixed reality

     very efficiently and in a short time.

     The scientific contribution of the documented project methodology is its unique-ness in the given  

     segment of the scientific discipline. From the available literature ana-lyzes and surveys it is not

     possible to find a more sophisticated projection methodology. The considerable advantage of this

     projection method based on the principles of mixed reality is confirmed by the verification of

     results in the real conditions of production practice. Time characteristics of its application when

     solving the deployment tasks of production technique of new, or of configured productions fully

     reflects the preserva-tion of the competitiveness (its increase) of the given production grouping in

     the di-mensions of quick time applicability and correction of risk factors of the application of the

     solution verified by simulation elements.

      The main contributions of the paper is the proposal of the methodical procedure for the design of   

      production structures using mixed reality based on the general prin-ciples of innovative design.

      Synthesis conditions for the implementation phase of de-signing production facilities in mixed

      reality and experimental use were defined. The method of typological relations was chosen for

      the implementation of proposals of the layout of production machines and robots and the

      selection of the optimal variant. The result of this method of project activity in spatial solutions of

      production systems is, for the necessities of practice, more accurate descriptions of the layout of

     machines in space using mathematical and other principles. Industrial grids enable to create im-

     mediate direct dimensional relationships between the exact surface and the location of the

     production system that realizes the production process, and the implementation of mixed reality

     allows to display a 3D model of the production equipment directly on the selected surface and

     immediately detect collision situations directly on the place and propose optimization of the

     location.

  1. Formal deficiencies and errors were removed in the whole paper.
  2. Chapter numbering was corrected.

5. The numbering of mathematical formulas was also corrected

Reviewer 2 Report

122- State the version of the app used for the experimental study

132- The research of two use cases revealed. Rewrite the statement properly. 

158-166 Move lines 158 to 166 from Introduction (section 1) to method in section 2 

170- List the materials used for your study

213- justify equation 2 to the right-hand side

218- The resolution of figure 1 is poor. Please improve on it.

237- List the conditions to be met to avoid collusion between the distributed construction elements. 

289- Results and Discussion is the most appropriate heading for section 3

405- Provide clear results in figure 8. You mention figure 8 before figure 7 in the results. That is not scientific.

Section 4 is missing in your manuscript.

536- Section 5 heading should be Conclusion and not Discussion and Conclusion. 

You did not discuss your results properly. The conclusion part of the manuscript is poor. Please rewrite 

Author Response

We corrected our paper according to your suggestions. All new aor corrected information are marked in yellow.

  1. Line 122 - ARKitTM and Unity3D versions have been added

              The app is made with the ARKitTM version 5.1.0 API tool and Unity3D 2022.1.0.

  1. Line 132 – the formulation of the text was supplemented and modified.

                The research paper written by Sautter and Dalling [20] looked at the benefits and drawbacks   

                of mixed reality-assisted on-the-job learning, and described some practical implications for

                industrial training. The research of two use cases (first - the training of new employees for   

                the semi-automated assembly lines in the production of pneumatic cylinders revealed that  

                mixed  reality, second - includes learning modules with a focus on declarativeand cognitive  

                knowledge)  technologies can assist learning factories overcome their constraints and  

                maximize their  potential for successful industrial training in key constructivist learning    

                areas.

  1. Lines 158 - 166 were moved from the “Introduction” chapter to the “Materials and Methods” chapter according to suggestion

      The proposed mixed reality structure consists of five layers: the first layer consid-ers system parts; 

      the second and third layers concentrate on architectural issues for component integration; the

      fourth layer is the application layer that executes the ar-chitecture; and the fifth layer is the user

      interface layer that allows user interaction [24]. During the COVID-19 pandemic, the suggested

      model with the usage of mixed re-ality provides a comprehensive solution for huge building

      complexes and industrial parks, guaranteeing public safety and also the health and well-being of

      the facilities management team. The mixed reality technique allows for the remote processing of

      path layouts, avoiding human interaction and ensuring that there is no chance of virus 

      transmission [25].

  1. Line 170 – the sources used in the study were added.

      The research link the techniques used in the educational process with the necessities of real ň

       practice and it is also proposed direct visualization solution for the operator in the machining

       process [26].

  1. Line 213 – the equation was aligned to the right margin
  2. Line 218 – we did not change the resolution of image 1, because is fine in our opinion.
  3. Line 237 – the conditions that must be taken into account in order to avoid collisions of production equipment have been added and described.

     (The geometric characteristics of individual production systems are also important for modelling

     the spatial relationships in the production systems. It is necessary to take into account spatial 

      relations and basic geometric characteristics: outer shape and dimensions, maximal dimensions of

     the working space, orientation and position of the coordinates in the coordinate system of the

     production element with respect to the reference coordinate system and geometric

     characteristics of the working space).

  1. Line 289 – the title was changed to “Result and Discussion”.
  2. Line 405 – deficiencies of image numbering in paper texts have been removed.
  3. Line 536 – the fourth chapter has been renamed to Conclusions
  4. The research results of the described method and the final part of the paper were added.

        Models of typological relationships in production systems are characterized by considerable   

        universality in terms of application conditions, as they are suitable for optimization of the 

        production system design, but also for the traditional design of production systems.

       The typological relationships presented in the paper are based on generally valid spatial  

       principles and methodologies that are obligatory when it is solved the layout of production 

       equipment on the production area. These are applied in decision-making software algorithms.

       Their output is the optimal solution that eliminates the conflict-ing states of the location of the 

       new means of production related to the immediate surroundings of the new layout, which is

       located through the Wumark brand and its optimal location is transformed into the mixed reality.

       When choosing the location of the Vumark´s, we used a projector to project the raster on the

       surface of the production hall, with which we determined the exact loca-tion of the production

       equipment and we eliminated collision situations when insert-ing the production equipment into 

       the layout of the production system.

       On the basis of the mentioned procedure, it is possible to place Vumark´s in the layout solution  

       within the optimization of production system and classic mechanical engineering productions for 

       piece and small batch production.

       The mentioned mathematical apparatus can be used in the placement of produc-tion equipment 

        in shapes that are often used in such types of production (e.g. type “U”, type “L”, type “I”, type 

       “diagonal” and type “star”).

Vumark's design method enables to make quick changes of various variants when placing machines in the already existing layout of the production system. Using this method, it is possible to immediately detect collision situations between a physical (re-al) machine and a virtual machine in production system and thus can avoid high costs and time losses during its physical installation. When verifying the placement of the virtual machine in the layout, it is not necessary to shut down the production process as a whole, which also eliminates the economic losses of the production plant. Virtual machine models can be obtained in software such as Factory Design, vis TABLE and Tarakos, or they can be modelled in a CAD system and implemented in mixed reality very efficiently and in a short time.

The scientific contribution of the documented project methodology is its unique-ness in the given segment of the scientific discipline. From the available literature ana-lyzes and surveys it is not possible to find a more sophisticated projection methodology. The considerable advantage of this projection method based on the principles of mixed reality is confirmed by the verification of results in the real conditions of production practice. Time characteristics of its application when solving the deployment tasks of production technique of new, or of configured productions fully reflects the preserva-tion of the competitiveness (its increase) of the given production grouping in the di-mensions of quick time applicability and correction of risk factors of the application of the solution verified by simulation elements.

The main contributions of the paper is the proposal of the methodical procedure for the design of production structures using mixed reality based on the general prin-ciples of innovative design. Synthesis conditions for the implementation phase of de-signing production facilities in mixed reality and experimental use were defined. The method of typological relations was chosen for the implementation of proposals of the layout of production machines and robots and the selection of the optimal variant. The result of this method of project activity in spatial solutions of production systems is, for the necessities of practice, more accurate descriptions of the layout of machines in space using mathematical and other principles. Industrial grids enable to create im-mediate direct dimensional relationships between the exact surface and the location of the production system that realizes the production process, and the implementation of mixed reality allows to display a 3D model of the production equipment directly on the selected surface and immediately detect collision situations directly on the place and propose optimization of the location.

Round 2

Reviewer 1 Report

A substantial part of the comments from review 1 have been accepted by the authors and the shortcomings have been corrected in the new version of the article. I would suggest adding to the article in view of comment 2 (The methodology for applying augmented reality for manufacturing layout design is described in a rather general way ...). I suggest adding in chapter 3 a description of the methodology by specifying the individual steps of the methodology or by showing the methodology and its individual steps in the form of a flowchart.

Author Response

All new pharagraphs are highlited in yellow.
